# Disulfide isomerization reactions in titin immunoglobulin domains enable a mode of protein elasticity

David Giganti[1], Kevin Yan[1], Carmen L. Badilla[1], Julio M. Fernandez[1] & Jorge Alegre-Cebollada [2]

The response of titin to mechanical forces is a major determinant of the function of the heart. When placed under a pulling force, the unstructured regions of titin uncoil while its immunoglobulin (Ig) domains unfold and extend. Using single-molecule atomic force microscopy, we show that disulfide isomerization reactions within Ig domains enable a third mechanism of titin elasticity. Oxidation of Ig domains leads to non-canonical disulfide bonds that stiffen titin while enabling force-triggered isomerization reactions to more extended states of the domains. Using sequence and structural analyses, we show that 21% of titin's I-band Ig domains contain a conserved cysteine triad that can engage in disulfide isomerization reactions. We propose that imbalance of the redox status of myocytes can have immediate consequences for the mechanical properties of the sarcomere via alterations of the oxidation state of titin domains.

[1] Department of Biological Sciences, Columbia University, New York, NY 10027, USA. [2] Centro Nacional de Investigaciones Cardiovasculares Carlos III (CNIC), 28029 Madrid, Spain. Correspondence and requests for materials should be addressed to J.A.-C. (email: jalegre@cnic.es)

Striated muscles share a similar organization of contractile filaments, in which the giant protein titin works as a molecular spring that sets the stiffness of myocytes during contraction and relaxation cycles[1]. In the extensible I-band portion of sarcomeres, the titin filament is composed of unstructured polypeptide regions and tandem immunoglobulin (Ig) domains (Fig. 1a). The mechanical properties of the I-band of titin arise from uncoiling/recoiling of the unstructured segments and unfolding/refolding of Ig domains[2–6]. Protein folding decreases the effective contour length of titin, leading to increased stiffness. In contrast, protein unfolding adds new residues to the unstructured pool of amino acids, making titin more elastic.

Titin-based myocardial stiffness is modulated by isoform switching and by posttranslational modifications, such as phosphorylation and S-glutathionylation[1,7–9]. It has also been proposed that oxidative stress can limit the elasticity of titin through intramolecular disulfide formation of the cardiac-specific, unstructured N2B region[10]. Disulfides are covalent bonds that cannot be cleaved by forces in the physiologically relevant pico-Newton (pN) range. Hence, disulfide bonds reduce the contour length of proteins, making them stiffer[11]. Similarly, engineered disulfides have been shown to accelerate mechanical folding of Ig domains, leading also to stiffer domains[11,12]. In this regard, the proposal by Mayans et al.[13] that disulfides could be present in several domains of the I-band of titin was very attractive. Using sequence similarity analyses, these authors proposed that 40% of the I-band Ig domains contain cysteines at positions that form the classical disulfide bond connecting β-strands B and F in extracellular Ig domains such as antibodies. However, deficient disulfide formation in vitro has so far precluded examination of the functional role of disulfides in titin domains[10,14].

Prompted by the availability of the crystal structure of segment I65–I70 of rabbit titin (pdb: 3B43[15]), which shows six Ig domains containing cysteines in geometries that are compatible with intradomain disulfide bond formation, we decided to investigate the mechanical effects of disulfides in the Ig domains of titin. We developed in vitro protocols to introduce disulfide bonds in I65–I70, and examined its mechanics using single-molecule atomic force microscopy (AFM) in force-clamp mode. Unexpectedly, we found that the preferred disulfide in domains that contain more than two cysteines is not the classical disulfide characteristic of extracellular Ig domains, but a titin-specific disulfide between β-strands B and G. This disulfide configuration enables mechanically induced isomerization reactions, which in turn allow fine-tuning of the response of titin to a stretching force.

## Results

**Oxidized titin Ig domains contain atypical disulfide bonds.** The crystal structure of rabbit I65–I70 reveals that all 6 domains contain at least two cysteines at equivalent positions in strands B, F and G (Fig. 1b, Supplementary Figure 1a,b). These cysteine residues, which we name CysB, CysF and CysG according to the β-strand they belong to, form a clustered triad in I67 and I69, or a pair in the other domains (Fig. 1c, Supplementary Figure 1c). The three types of cysteines are buried in the hydrophobic core of the domains and their thiol groups are oriented towards each other and co-localized within a 6 Å sphere. Their close proximity suggests that the cysteines can be involved in disulfide formation. Propensity to disulfide bond formation is suggested by the presence of a CysB–CysG bond in domain I69 despite the unfavorable conditions used to produce and crystalize the recombinant I65–I70 fragment[15]. Human I65–I70 (residues 7945–8511 in uniprot Q8WZ42-1) is 95% identical to rabbit I65–I70 and all 14 cysteines are conserved (Supplementary Note 1).

To examine the effect of disulfides on the mechanical properties of titin domains, we employed single-molecule force spectroscopy by AFM at constant force (force-clamp). Mechanical unfolding trajectories of single polyproteins unambiguously reveal the existence and position of disulfide bonds that cannot be cleaved by force, resulting in limited polypeptide extension under force[11,12,16]. We expressed rabbit I65–I70 in *Escherichia coli* and applied a diluted solution in the AFM (Fig. 2a). When pulled at 170 pN, the I65–I70 substrate unfolds and produces traces with a step-wise extension of the tethered polyprotein in which every step marks the mechanical unfolding of a domain in the I65–I70 protein[17]. All steps display a similar length with a normal distribution centered at $26.4 \pm 0.7$ nm ($N = 496$), which corresponds to the predicted mechanical extension of single disulfide-free Ig domains (Fig. 2b, c, Supplementary Note 2). Shorter

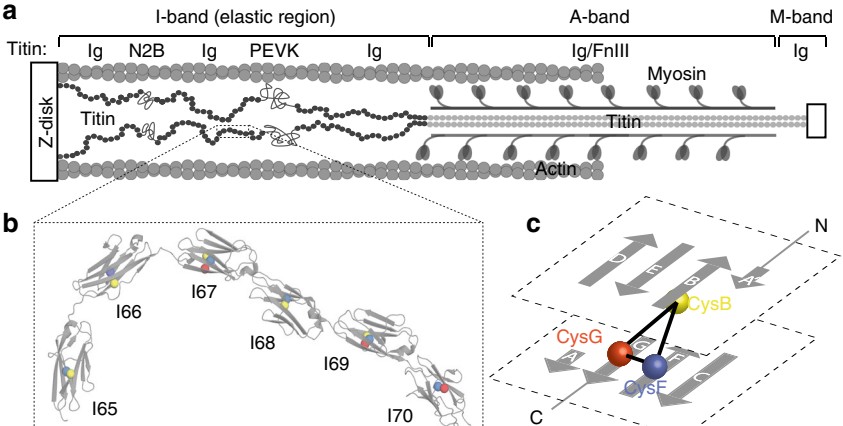

**Fig. 1** Conserved cysteine positions in the I65–I70 fragment of titin. **a** Schematic architecture of one half of the sarcomere. The titin filament spans across the contractile unit and its elastic properties arise from its extensible I-band region where Ig modules (filled black circles) can mechanically unfold and refold. The two entropic springs N2B and PEVK are also depicted as unstructured chains. Gray circles represent the repetition of Ig and fibronectin domains (FnIII) in the A- and M-band. **b** Crystal structure of the rabbit titin segment I65–I70 (pdb: 3B43). This 566 amino acid-long portion of proximal titin is organized in six tandem Ig modules represented in gray. Cysteines are marked by spheres, which are colored according to their position in the Ig secondary structure topology. Cysteines in β-strands B, F and G are shown in yellow, blue and red, respectively. **c** Scheme of the topology of a typical Ig domain of titin[50], which displays the arrangement of the three clustered cysteines CysB, CysF and CysG. CysF and CysG appear in two adjacent β-strands whereas CysB is located in the opposite β-sheet (upper plane)

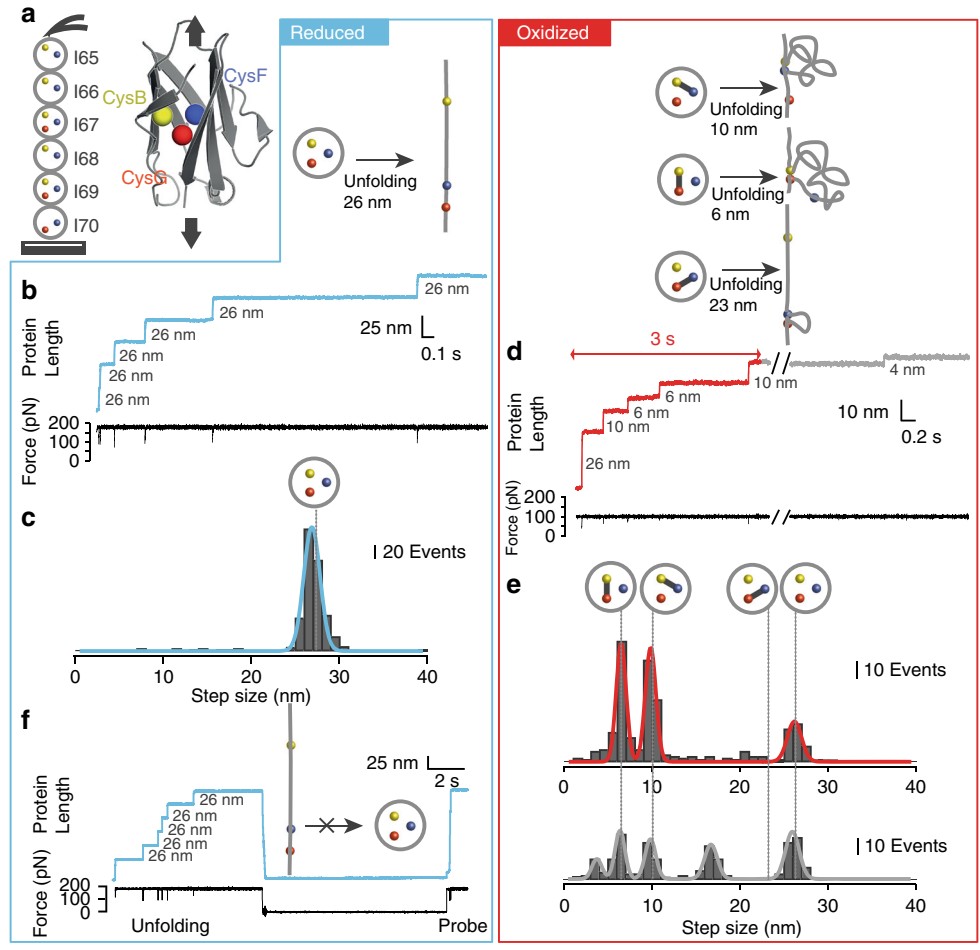

**Fig. 2** Disulfide bonds in I65–I70. **a** Schematic representation of cysteine position in the six Ig domains composing I65–I70, and X-ray structure of I67 (pdb: 3B43) with arrows representing the applied force at the termini in AFM experiments. CysB, CysF and CysG are highlighted. Step sizes are indicated in the diagrams. We fitted Gaussian curves to all histograms presented here to estimate the different populations of observed step sizes. **b** Typical trace recorded applying a constant force of 170 pN on I65–I70$_{reduced}$. **c** The histogram of step sizes shows that only reduced domains are detected (unfolding step of 26 nm). **d** Stretching I65–I70$_{oxidized}$ at 100 pN produces new distinct populations of shorter steps, with a different kinetics of appearance. The first 3 s of the experiment are shown in red. In this experimental recording, the polyprotein extends over time in steps of 26, 10, 6, 6 and 10 nm and in a last 4 nm step (in light gray, a 15 s portion of the trace has been cut out). **e** Histograms of the step sizes obtained with I65–I70$_{oxidized}$ (red in the first 3 s and gray for delayed events). The I65–I70$_{oxidized}$ unfolds showing two large populations of 6 and 10 nm steps in agreement with the unfolding and extension of domains shortened by disulfides CysB–CysG and CysB–CysF, respectively. The presence of longer 26 nm steps illustrates that some domains remain reduced. The steps at 4 and 17 nm correspond to events of disulfide isomerization. **f** This experimental trace obtained with a typical three-pulse protocol allows the evaluation of the refolding ability of I65–I70$_{reduced}$. All domains mechanically unfold during a first pulse (unfolding pulse), then collapse when the force is switched off for a set time Δt (quench). The third pulse (probe) probes the domains that recovered their native state during the quench phase. In this trace, the absence of steps in the probe pulse indicates that none of the mechanically unfolded domains refolded during the quench (Δt = 10 s)

extensions would be expected if domains containing disulfide bonds unfold. Hence, our standard purification procedure renders fully reduced I65–I70 domains, in agreement with the expression of the protein in the reducing environment of the cytoplasm of *E. coli*[18]. Kinetics of unfolding shows that the mechanical stability of domains I65–I70 is similar to other titin modules, in agreement with previous studies on human I65–I70[2,19].

Even when in close proximity, disulfide linkage between two neighboring cysteines is extremely slow and requires electron acceptors[20,21]. Storage of I65–I70 for 2 weeks at 4 °C did not increase the proportion of shorter unfolding steps coming from disulfide-containing domains. Slightly better oxidation yield was obtained by incubation with 0.3% H$_2$O$_2$ for 72 h, which is a efficient oxidation protocol for exposed cysteine residues[10]. Hence, disulfide bond formation appears unfavorable once the Ig domains in I65–I70 are folded, suggesting that the cysteines are

buried and not accessible to oxidants in solution[3]. We found that incubations at higher temperatures and in the presence of CuCl$_2$ improved oxidation yields. Under the optimized oxidation conditions, only 22% of the AFM steps correspond to the mechanical unfolding of reduced domains (26 nm steps). The remaining steps are shorter due to the presence of disulfide crosslinks that block mechanical extension of the amino acids trapped behind the disulfides[11] (see below and also Supplementary Note 2 for assignment of step sizes). Hence, our optimized protocols are able to efficiently introduce disulfide bonds in the majority of the Ig domains in I65–I70.

In the recording shown in Fig. 2d, a single I65–I70$_{oxidized}$ polyprotein is stretched at a constant force of 100 pN, producing six discernable steps of 26, 10, 6, 6, 10 and 4 nm. We collected several equivalent traces that contain at least two steps of the same size and found that an important fraction of the shorter steps emerges with a faster time course than the 26 nm steps

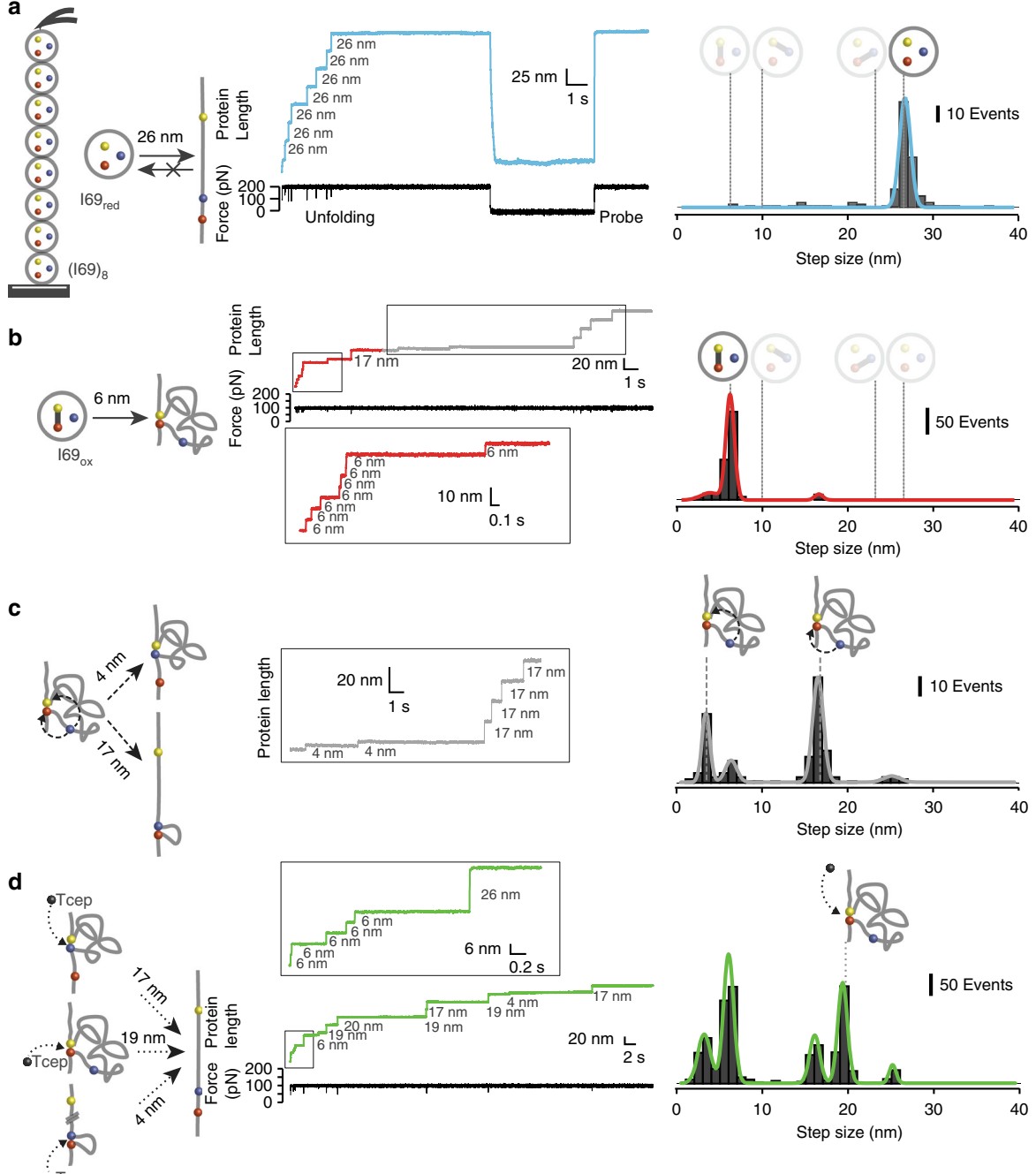

**Fig. 3** Mechanical characterization of I69_oxidized identifies disulfide isomerization as a contributor to titin elasticity. Diagrams on the left describe the mechanism of extension of I69 due to unfolding **a**, **b** and unfolding-triggered redox reactions **c**, **d**. Each pathway is illustrated with a trace and a histogram of step sizes. Theoretical step sizes are indicated by dashed lines (Supplementary Note 2). **a** The unfolding of (I69_reduced)_8 pulled at 170 pN results in a uniform step-wise extension of the polyprotein (26 nm steps in the blue trace). Similar to the trace shown in Fig. 2f, the absence of steps in the probe pulse indicates low refolding kinetics of I69_reduced ($\Delta t = 5$ s). **b** When (I69_oxidized)_8 is stretched at 100 pN, a predominant population of 6 nm emerges in the first 3 s, marking the unfolding of I69 domains containing disulfide CysB–CysG. **c** After 3 s (gray part of the trace in (**b**)), two new populations of 4 and 17 nm steps appear. Mechanical unfolding activates a reaction of disulfide exchange where the thiol group of CysF can attack either the sulfur of CysB or CysG, giving rise to the 4 and 17 nm step populations. **d** In the presence of 10 mM Tcep, additional 19 nm steps are detected (Gaussian fit centered at $19.2 \pm 0.3$ nm), marking the reduction of disulfide CysB–CysG by Tcep (black sphere). The reduction of the newly interchanged disulfide CysB–CysF and CysF–CysG along the trajectory produces steps of 3 and 16 nm[22]

observed when reduced I65–I70 domains unfold (Fig. 2e, red color marks events happening in less than 3 s). Gaussian fits to the global histogram of step sizes gives 5 peaks of $3.6 \pm 0.4$, $6.5 \pm 0.4$, $9.8 \pm 0.5$, $16.7 \pm 0.4$ and $26.1 \pm 0.6$ nm. Among them, the 10 and 6 nm steps match the theoretical shortened extension of

domains with classical disulfide CysB–CysF, and the atypical disulfide CysB–CysG, respectively (Supplementary Note 2). In contrast, the absence of 23 nm steps suggests that CysF–CysG disulfide is not favored under our experimental conditions. While the relative frequency of the 3 disulfides gives some indications

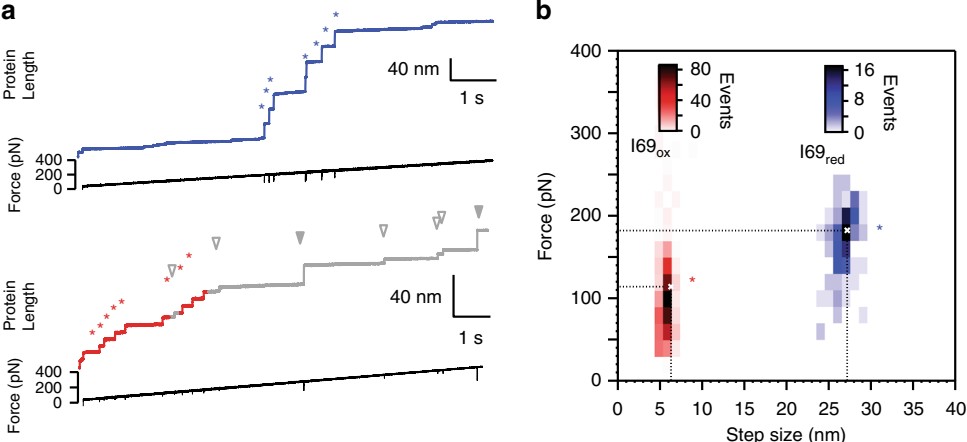

**Fig. 4** Disulfide bonds decrease the mechanical stability of titin Ig domains. **a** Typical unfolding traces of $(I69)_8$ recorded with a linear force increase of 40 pN s$^{-1}$. Blue asterisks mark the unfolding events of $I69_{reduced}$ (26 nm steps in the blue upper trace) whereas red asterisks correspond to unfolding steps of $I69_{oxidized}$ (6 nm steps in the red lower trace). Isomerization reactions are detected as additional steps (gray triangles). **b** Unfolding forces and step sizes are represented in a bidimensional histogram. Average values appear as white crosses

about the oxidation state of the I65–I70 hexamer, the precise arrangement of disulfide bonds in I65–I70$_{oxidized}$ cannot be assigned since there are multiple disulfide combinations compatible with experimental results. In addition, the slower events marked by 4 and 17 nm steps (Fig. 2e, gray histogram) cannot be explained by mechanical unfolding events.

**Non-classical disulfides enable force-induced isomerization**. To avoid the heterogeneity of I65–I70, we studied the $(I69)_8$ homopolyprotein, which only contains domain I69 and therefore allows unambiguous identification of disulfides. I69 contains all CysB, CysF and CysG, and could form three different disulfides. Similar to I65–I70, $(I69)_8$ is expressed in standard conditions as a fully reduced protein that shows an average mechanical unfolding step size of 26 nm ($N = 136$, Fig. 3a). Under oxidizing conditions, $(I69)_8$ is oxidized virtually to completion, as demonstrated by the rare appearance of 26 nm unfolding steps when I69$_{oxidized}$ is pulled in the AFM (Fig. 3b, c). Hence, I69 seems more prone to oxidation than other domains of I65–I70, in agreement with the observation that I69 is the only domain that is oxidized in the crystal structure of I65–I70[15]. The mechanical unfolding of I69$_{oxidized}$ produces fast length increases of 6 nm (Fig. 3b), which mark specifically the unfolding of domains containing disulfide CysB–CysG (Supplementary Note 2). No events corresponding to the unfolding of I69 containing other disulfide combinations were detected (10 or 23 nm steps, Supplementary Note 2). Hence, our pulling experiments define the non-classical CysB–CysG as the only disulfide present in I69$_{oxidized}$. Interestingly, the same CysB–CysG disulfide is present in the crystal structure of I65–I70[15].

Cysteines in loops delimited by disulfide bonds can engage in mechanically activated thiol/disulfide exchange reactions leading to disulfide isomerization[22,23]. Upon mechanical unfolding of I69$_{oxidized}$, CysF is located in the loop connecting CysB and CysG, in a geometry that is compatible with isomerization reactions (diagram in Fig. 3c). Accordingly, 6 nm steps coming from unfolding of I69$_{oxidized}$ are followed by delayed steps of 4 and 17 nm (Fig. 3b, c). These slow steps match the expected size resulting from disulfide isomerization reactions where CysF attacks on CysB to produce disulfide CysB–CysF (releasing 11 amino acids that cause a 4 nm step) or on CysG to generate CysF–CysG (releasing 51 amino acids that lead to a 17 nm step) (diagram in Fig. 3c, Supplementary Note 2). In 44 AFM traces that only contain unfolding and isomerization events, we found that there

is always at least one 6 nm unfolding step preceding one 4 or 17 nm isomerization step, which gives additional support to our assignment of steps to unfolding or isomerization reactions (Supplementary Figure 2). These results using I69$_{oxidized}$ strongly suggest that the 4 and 17 nm steps detected when pulling from I65–I70 (Fig. 2e) arise from isomerization reactions involving domains I67 and I69.

The relative frequency of 4 and 17 nm steps defines the regiospecificity of the nucleophilic attack of CysF on disulfide CysB–CysG[22]. We measured the dwell time of each type of event and developed a kinetic model that accounts for the unfolding rate of I69$_{oxidized}$ (1.06 s$^{-1}$), and the speed of appearance of the 4 and 17 nm steps at 100 pN (Supplementary Figure 2). We obtain the following rates at 100 pN: $k_{iso1} = 0.021$ s$^{-1}$ and $k_{iso2} = 0.027$ s$^{-1}$ (Supplementary Figure 3). Hence, the regiospecificity $k_{iso2}/k_{iso1}$ is 1.29, i.e., the formation of disulfide CysF–CysG, which results in the longer 17 nm extension, is moderately favored. As expected, a similar regioselectivity (1.3) is obtained from the ratio of detected 4 and 17 nm steps.

**Disulfide CysB–CysG is buried in the core of titin domains**. In order for disulfide isomerization to be an effective contributor to the elasticity of titin, disulfide CysB–CysG must be stable in the reducing environment of myocyte cytosol when domains are folded. According to the crystal structure of I65–I70, the CysB–CysG bond in I69 appears buried in the hydrophobic core, and therefore not accessible to reducing metabolites. To confirm this observation, we pulled I69$_{oxidized}$ in the presence of the reducing agent Tcep. We find that I69 domains remain oxidized and extend with the same 6 nm steps, even after incubation with 10 mM Tcep for 1–4 h, demonstrating that disulfide CysB–CysG remains inaccessible to the solvent in the folded state of I69 (Fig. 3d, inset)[16,24].

In these pulling experiments in the presence of Tcep, a new population of 19 nm steps emerges with a similar time course than isomerization events (Fig. 3d, Supplementary Figure 2c). The 19 nm steps match the expected length resulting from the intermolecular cleavage of disulfide CysB–CysG by Tcep (diagram in Fig. 3d, Supplementary Note 2). Hence, mechanical unfolding of I69$_{oxidized}$ in the presence of Tcep results in steps corresponding to oxidized protein unfolding (6 nm), isomerization of disulfide CysB–CysG (4 nm and 17 nm) and intermolecular reduction of disulfide CysB–CysG by Tcep (19 nm). In the experiments where Tcep is present, we also expect a fraction of

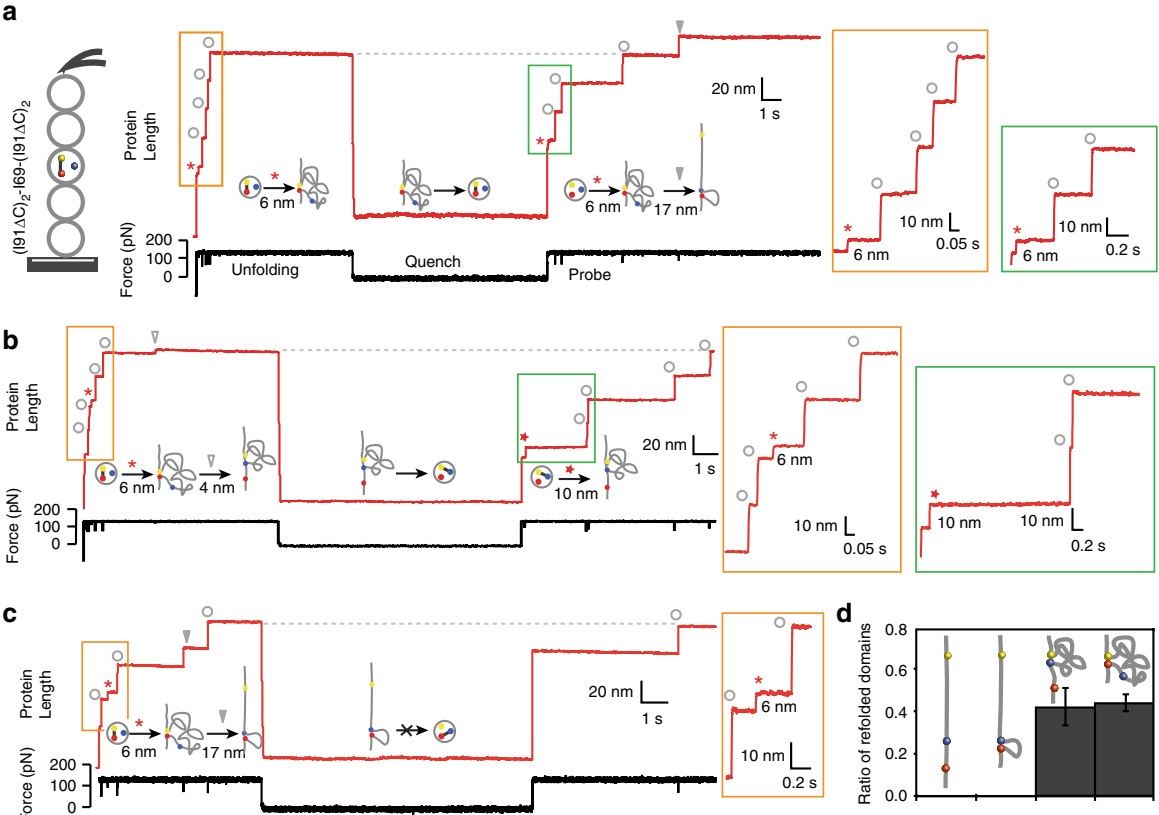

**Fig. 5** Disulfide bonds enable refolding of I69. **a** We use the engineered heteropolyprotein (I91ΔCys)$_2$–I69-(I91ΔCys)$_2$ to characterize the influence of disulfide bonds on the refolding of I69$_{oxidized}$ (quench Δt = 10 s, unfolding/probe force = 130 pN). Only traces that show the same extension (doted gray lines) at the end of the unfolding and probe pulses (before isomerization steps in the probe pulse) are considered. This recording indicates that I69$_{oxidized}$ unfolds in the unfolding pulse (6 nm step) and refolds during the quench while the disulfide CysB–CysG remains formed (6 nm step in the probe pulse). In this example trace, a subsequent isomerization step is detected in the probe pulse (solid triangle). **b** In this trace, disulfide CysB–CysF is formed through isomerization during the unfolding pulse (4 nm step, gray empty triangle). A 10 nm step in the probe indicates that I69 has refolded during the quench pulse keeping disulfide CysB–CysF. **c** Alternatively, if CysF–CysG forms (17 nm step, gray filled triangle), the absence of 23 nm step in the probe suggests an inability to refold during the quench. **d** Histogram summarizing how the three possible disulfide bonds formed within the triad affect the refolding probability of I69$_{oxidized}$ for a quench time of 10 s, as determined with the protein (I91ΔCys)$_2$-I69-(I91ΔCys)$_2$. Data for I69$_{reduced}$ were obtained with (I69)$_8$. The refolding probability and associated SEM represented by the error bars were calculated by bootstrapping. N = 131 (CysB–CysG), N = 31 (CysB–CysF) and N = 38 (CysF–CysG)

the 4 and 17 nm steps to correspond to reduction of newly isomerized disulfides CysB–CysF and CysF–CysG by Tcep, which produce steps of similar size as the isomerization reactions, making it challenging to discriminate between them (diagram in Fig. 3d, Supplementary Note 2)[22,23].

**Disulfide CysB–CysG mechanically weakens Ig domains**. When the titin filament is pulled during myocyte contractile activity, around 100 Ig domains of the I-band of titin experience end-to-end force. Mechanical unfolding of only a few of them can be enough to reduce passive tension[2]. In this scenario, domains with a lower mechanical stability are primed for mechanical unfolding and prevent further Ig unfolding. Hence, to estimate the extent of disulfide isomerization reactions in titin, we need to consider how often oxidized domains visit the unfolded state relative to reduced domains. To measure how disulfide CysB–CysG affects mechanical stability, we pulled I69 polyproteins using a linear increase in force of 40 pN s$^{-1}$. We found that oxidized domains unfold earlier in the ramp (Fig. 4a). Under our conditions, the average unfolding force of I69$_{reduced}$ domains is 182 pN (N = 134) while I69$_{oxidized}$ unfolds at an average force of 114 pN (N = 534) (Fig. 4b). Fitting our data to Bell's model shows that the distance to the transition state, which determines the force sensitivity of

the unfolding reaction, is similar for oxidized and reduced I69 domains (Supplementary Figure 4)[25]. Hence, we conclude that oxidized I69 domains are more prone to unfolding than reduced counterparts at any force.

**Disulfide bonds accelerate folding of titin domains**. During contraction–relaxation cycles of myofibrils, refolding of Ig modules maintains titin-based stiffness[2,3]. In the case of disulfide-containing domains, mechanical unfolding enables intermolecular reduction events by exposing disulfides to the solution. Hence, the speed at which disulfide-containing domains are able to refold is a key factor controlling the endurance of titin disulfides in the reducing environment of the cytosol.

We measured the folding properties of titin domains by subjecting polyproteins to an unfolding pulse, followed by a quench pulse to 0 pN. We monitored refolding by pulling again to high force (probe pulse). Single-molecule events are fingerprinted by the same extension of the polyprotein at the end of the unfolding and probe pulses[26]. Domains that unfold in the probe pulse mark those ones that managed to refold during the quench time. The ratio between the number of unfolding events observed in the probe pulse and those observed in the unfolding pulse reports the refolding fraction for a given quench time[3]. It has

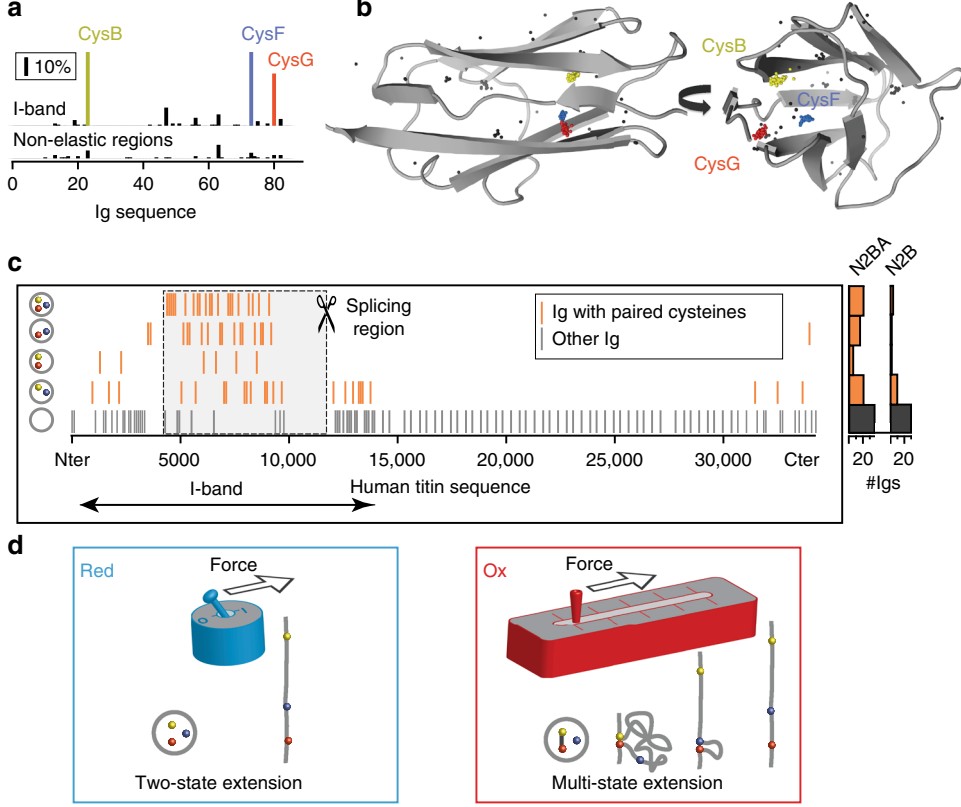

**Fig. 6** Disulfide isomerization in the Ig domains of titin. **a** Conservation of cysteines in the 163 Ig domains of the canonical titin. The upper histogram displays the cysteine percent conservation in the 101 Ig modules located in the I-band of titin. The lower histogram presents the percent conservation of cysteines in the 62 Igs in non-elastic regions (Z-Disk, A-Band and M-Band). The most conserved cysteines CysB, CysF and CysG, are shown in yellow, blue and red, respectively, and are over represented specifically in the I-band. **b** Mapping of cysteines in the structure of Ig domains of the I-band of titin. This 3D map is obtained by superimposition of the 101 Ig structures (X-ray and homology models) belonging to the I-band. For clarity, only the structure of I91 is represented. Each cysteine is marked at its Cβ atom by a colored dot for the cysteines CysB (yellow, C23'), CysF (blue, C73') and CysG (red, C80') or a black dot for other cysteine positions. **c** Left: Each bar marks the position along the canonical titin sequence of domains with specific distributions of CysB, CysF and CysG. These domains are concentrated in the I-band. Cardiac titin filaments are expressed in two major isoforms, the long N2BA (3.7 MDa) and the shorter N2B (2.97 MDa)[28,51,52]. The section highlighted by the dashed line, which spans between exons 50 and 219, is highly spliced and totally absent in the N2B isoform. Right: The histogram displays the change of the relative populations of the different domains in the two major isoforms of cardiac titin (see also Supplementary Note 6). **d** Model of extensibility of Ig domains in the I-band. Mechanical unfolding of disulfide-free Ig modules ("Red") results in full extension of the polypeptide. In contrast, the mechanical extension of oxidized Ig domains containing the cysteine triad ("Ox") is also determined by isomerization and reduction reactions. We propose that disulfide bonds in titin act as a redox-sensitive mechanical potentiometer that can finely tune titin's elasticity

been observed before that Ig modules from titin refold efficiently. Indeed, the proximal I4 domain and the well-characterized distal I91, both located in the I-band, refold with rates of 0.3 s$^{-1}$ and 1.2 s$^{-1}$, respectively[2,27]. In contrast, we only detected rare refolding events of I65–I70$_{reduced}$ (refolding fraction = 0.07 ± 0.02 for a 5 s quench time, N = 183) (Fig. 2f). Similarly, low refolding ability was observed for I69$_{reduced}$ (refolding fraction = 0 for a quench of 5 s and 10 s, N = 88 and N = 120, respectively) (Fig. 3a). I69$_{reduced}$ also failed to refold at longer quenching times of up to 30 s (Supplementary Figure 5a). These results indicate a significantly slower folding rate of reduced I69 and most of other domains in I65–I70, by at least 1 order of magnitude compared to I4 or I91.

To determine the effect of disulfides in the folding properties of titin domains, we repeated the unfolding/quench/probe experiments using (I69$_{oxidized}$)$_8$. However, these experiments bring new challenges to the identification of single-molecule events due to isomerization reactions that modify the contour length of the protein (Supplementary Figure 5b). To avoid the complexity brought by multiple isomerization reactions, we constructed the heteropolyprotein (I91ΔCys)$_2$-I69-(I91ΔCys)$_2$ that contains only 3 cysteines in the central I69 domain and is fingerprinted by the

cysteine-free I91 modules, whose mechanical properties are equivalent to wild-type I91[3]. This construct enabled us to monitor the disulfide status of domain I69 in fingerprinted single-molecule folding traces. In the typical recordings obtained with (I91ΔCys)$_2$-I69$_{oxidized}$-(I91ΔCys)$_2$ in the absence of isomerization, one I69$_{oxidized}$ domain unfolds producing a 6 nm step, while the fingerprinting I91ΔCys unfolding is marked by distinct steps of 25 nm (Fig. 5a). These fingerprinting traces reach the fully extended length of the polyprotein after refolding, and afterwards may show one additional step indicating isomerization of disulfide CysB–CysG as in the example trace in Fig. 5a (17 nm step in the probe pulse, indicated by a solid triangle). These traces show that after a quench of 10 s the refolding fraction of I69 containing disulfide CysB–CysG is 0.44 ± 0.04 (N = 131) (Fig. 5d, Supplementary Figure 6).

Experiments with (I91ΔCys)$_2$-I69-(I91ΔCys)$_2$ also allowed us to explore the contribution of disulfides CysB–CysF and CysF–CysG to I69 refolding. With this aim, we focused on single-molecule traces where the unfolding pulse contains the signature of specific isomerization of CysB–CysG to CysB–CysF (4 nm step, such as the trace in Fig. 5b) or CysF–CysG (17 nm

step, such as the trace in Fig. 5c), and determined the refolding fraction of domains containing the isomerized disulfides (see also Supplementary Figure 6). In these recordings, refolded domains unfold producing the characteristic step size of the disulfide present at the end of the unfolding pulse (10 nm for CysB–CysF disulfide, Fig. 5b and Supplementary Figure 6), which gives extra confidence in the identification of refolding events. Results reveal that disulfide CysB–CysF also accelerates the folding of I69 reaching a refolding fraction of $0.42 \pm 0.09$ (10 s quench, $N = 31$) (Fig. 5d). In contrast, no refolding was observed in domains containing disulfide CysF–CysG (refolding fraction = 0, 10 s quench, $N = 38$). Overall, our results show that disulfide bonds in titin domains greatly accelerate protein folding.

**Over 20% of titin's I-band domains sustain isomerization.** To assess the relevance of disulfide isomerization at the level of the whole titin molecule, we turned to bioinformatics analyses. The giant titin gene is expressed in various transcripts of different lengths. Despite its large size and complexity, the structural organization of tandem repeat domains is well annotated[28]. The canonical sequence corresponding to the long N2BA cardiac isoform (uniprot Q8WZ42-1) contains 34,350 amino acids arranged in series of 163 Ig and 132 fibronectin domains. In the N2BA variant, titin contains 459 cysteines. All cysteines in the I-band, except for six cysteines that belong to the unstructured N2B region[10], are located in Ig domains. There are no cysteines in the PEVK unstructured region or the interdomain linkers in the I-band. Titin's I-band Ig domains can contain up to 5 cysteines, whereas cysteines become less common in the Ig domains of the Z-disk, A- and M-bands (Supplementary Figure 7).

To explore whether cysteine residues appear at conserved locations in the Ig domains of titin, we aligned the sequences of all 163 titin's Ig domains. The multiple alignment shows that 5 cysteines are remarkably conserved (Supplementary Note 3). Other cysteines, such as the rare cysteines that form a disulfide in the crystallographic structure of I1[13], appear in less than 5% of the domains. Here, we number cysteines according to the corresponding position in the alignment of the I91 domain (pdb: 1TIT) and use the nomenclature Cxx' to refer to them (Supplementary Note 4).

The conserved cysteines C47' and C63' are distant in the Ig fold and cannot form a disulfide bond between them in folded domains[12] (Supplementary Figure 8). The 3 other conserved cysteines C23', C73' and C80' are the most abundant and are present in 43, 42 and 30% of Ig domains, respectively, mainly in the I-band (Fig. 6a). These cysteines correspond to CysB, CysF and CysG in I65–I70. Hence, the fragment I65–I70 is an illustrative example of most disulfide-containing Ig domains in titin. We extended our observations to the remaining domains of titin using homology models[3]. Mapping all cysteine residues in the structure of all Ig domains highlights a well-defined cluster formed by CysB, CysF and CysG, at distances compatible with disulfide bond formation (Fig. 6b, Supplementary Figures 1c, 8, 9). Interestingly, CysB, CysF and CysG appear frequently in pairs or as a triad in the Ig domains. Out of 101 Ig domains in the I-band of the canonical N2BA titin, 43 Igs contain one of the three possible pairs, 21 Igs contain altogether the three cysteines and 24 Igs display a single cysteine out of the CysB, CysF and CysG triad (Fig. 6c). Hence, in 21% of domains of titin I-band domains, there is the possibility of disulfide isomerization reactions involving the triad CysB–CysF–CysG.

## Discussion

Here, we show that an extensive collection of cysteines in the giant myofilament protein titin can form intradomain disulfide bonds. The cysteine triad formed by CysB, CysF and CysG is very conserved in the Ig domains of titin. Indeed, the triad emerges as a widespread feature of titin Ig domains (Supplementary Note 5). In other protein families, many examples of clustered cysteines, often associated with histidine residues, participate in the tetrahedral coordination of metals[29]. However, no metal was observed in the crystal structure of I65–I70 and there are no neighboring histidines in the hydrophobic core of the domains. Instead, the triad enables isomerization reactions that directly modulate titin's mechanical properties.

In I65–I70, CysB and CysG are found together only in the triad-containing domains I67 and I69 (Fig. 2a, Supplementary Figure 1). Since disulfide CysB–CysG in I65–I70$_{oxidized}$ is detected with high frequency (6 nm unfolding steps, Fig. 2d, e), it is conceivable that I67 also forms the same unique CysB–CysG disulfide. Therefore, our results suggest that in the 21% I-band titin domains that contain the cysteine triad CysB, CysF and CysG, the preferred disulfide is established between CysB and CysG. This atypical CysB–CysG disulfide, different from the classical disulfide in extracellular Ig proteins, endows titin domains with an intrinsic two-step mechanism of extensibility under force, in which mechanical unfolding does not result in all-or-none extension as in reduced domains (Fig. 6d). Instead, full mechanical extension of oxidized triad-containing domains requires two subsequent reactions, i.e., unfolding and isomerization, which is a mode of protein elasticity that contributes to the overall mechanical properties of titin. A potential limitation of our study is that we induced oxidation in vitro, which may bias distribution of obtained disulfides with respect to native disulfides. However, the I69 domain in I65–I70 (pdb: 3B43) shows a spontaneously established CysB–CysG disulfide, which supports that in vitro oxidation captures a natural tendency of triad-containing domains to form disulfide CysB–CysG.

Disulfide isomerization reactions involve deprotonation of the free thiol and subsequent nucleophilic attack on one of the sulfur atoms in the disulfide[30]. In the folded structure of I69, CysF is near both sulfur atoms establishing the CysB–CysG disulfide (Supplementary Figure 1c). Hence, CysF may trigger isomerization reactions while I69 is folded, which would lead to disulfides CysB–CysF or CysF–CysG. Despite such spatial proximity, our single-molecule recordings do not show any evidence of those disulfides in I69$_{oxidized}$ (marked by 10 nm or 23 nm unfolding steps, respectively, Fig. 3b) suggesting that isomerization in the folded state is very slow, or that both CysB–CysF and CysF–CysG disulfides are very unstable and rapidly revert to CysB–CysG. We support the former possibility since the local hydrophobic environment of the three cysteines possibly favors the inert and protonated thiol form preventing disulfide rearrangements once in the folded structure. In addition, attaining the right geometry required for the nucleophilic attack is probably difficult in the context of a rigid folded domain[31,32]. In summary, the chemical and/or geometrical restraints of CysF in the folded state seem to hinder the nucleophilic attack needed for isomerization of disulfide CysB–CysG whereas isomerization can readily occur after mechanical unfolding.

Previous reports show that linked cysteines in Ig domains can alter the mechanical stability of the parent domain in a position-dependent manner[11,33–35]. Our results demonstrate that native disulfide bonds in titin mechanically weaken the domains. Interestingly, the superimposition of the three-dimensional (3D) structure of I69, oxidized in the crystal, with the other five reduced domains of I65–I70, shows a perfect conservation of the backbone structure (root mean square deviation < 2 Å). In all domains, there is no noticeable change in the structure of the mechanical clamp motif established by the hydrogen bonds between β-strands A' and G, which is responsible for mechanical

stability of Ig folds (Supplementary Figure 1d)[36]. Hence, we hypothesize that a more global effect induced by the disulfide in the force-stressed protein can decrease the mechanical stability of the domains. The fast time course of appearance of the 10 nm events during the extension of I65–I70$_{oxidized}$ suggests that disulfide CysB–CysF also lowers mechanical stability (Fig. 2d, e).

In our single-molecule experiments, we also detect that disulfides greatly accelerate mechanical folding of titin domains (Fig. 5d). This effect adds up to the increased ability of disulfide-containing domains to undergo hydrophobic collapse, which makes them more competent to fold[37]. We speculate that fast refolding can be important to limit the time disulfides are exposed to the reducing environment of the cytosol, enabling efficient "cocking" of disulfide-containing domains during the contraction phase of muscle activity. Together with the lower mechanical stability of disulfide-containing domains, our results suggest that unfolding of disulfide-containing domains is favored over reduced domains. Hence, as the titin filament stretches, unfolding of disulfide-containing domains ensures further extensibility by disulfide isomerization and reduction reactions, which in turn set the folding properties of titin domains during contraction[4]. Due to their low refolding rate, the newly generated reduced domains are expected to remain unfolded for long times, which may contribute to the ability of chaperones to bind the I-band of titin under mechanical stress (Figs 2f, 3a and 5)[38]. To support the generality of our observations, we have done clustering analysis based on the sequence of the Ig domains of titin. We have found that I65–I70 falls into a conservation cluster that contains 51 out of 58 domains in the differentially spliced region of titin (Supplementary Figure 10).

Titin is found in all striated muscles. Several titin isoforms result from differentially spliced mRNA transcripts[39]. It is known that titin passive tension is directly linked to isoform size, so changes in isoform allow fine-tuning of titin's function to the wide types of muscles in vertebrates. For instance, titin co-expresses in the heart as two main isoforms, N2BA and N2B, that differ in their size and mechanical properties[1]. In Fig. 6c, we show that the canonical cardiac N2BA titin contains the 4 types of cysteine combinations that can give rise to disulfide bonding based on the triad CysB, CysF and CysG. 21 domains contain the full triad, whereas 24, 17 and 6 domains have only cysteine pairs CysB–CysF, CysF–CysG and CysB–CysG, respectively. Triad-containing domains are enriched in the proximal part of the I-band, which corresponds to the highly spliced region of titin[8,40]. Hence, alternative splicing modulates the relative proportion of disulfide-isomerization-capable domains in the elastic I-band portion of titin (21% of Ig domains in N2BA titin and 9% in N2B titin). In long titin isoforms, the proportion of disulfide-containing domains in the I-band exceeds 60% (Fig. 6c and Supplementary Note 6).

To explore the overall effect of disulfide formation and isomerization in the mechanical properties of titin, we have adapted kinetic Monte Carlo simulations previously used to examine the molecular origin and regulation of the elasticity of titin (Supplementary Figure 11)[2,3]. In our simulations, we impose a cyclic force to the I-band of titin and monitor the resulting length, which depends on the entropic extension of both N2B and PEVK springs, the length of folded Ig domains and the stochastic unfolding, refolding and isomerization reactions involving Ig domains. Our simulations consider the precise arrangement of cysteines in the Ig domain structures (Fig. 6c), as well as experimentally determined rates of unfolding, refolding and disulfide isomerization of cysteine-containing titin domains (Fig. 5, Supplementary Figures 2-4). We find that titin fluctuates around a steady-state length after ~5 min under a 1 Hz triangular force cycles between 0 and 30 pN. This behavior is due to the

completion of disulfide isomerization reactions and to similar probabilities of protein unfolding and folding (Supplementary Figure 11b). The steady-state length at 30 pN is on average ~200 nm shorter in the N2BA isoform when disulfides are present in Ig domains. Hence, our simulations show that disulfide bond formation in titin domains leads to global stiffening of the titin filament[10]. Such stiffening is less apparent in N2B titin in agreement with the lower proportion of disulfide-containing domains in this isoform, and it is slightly more prominent in simulations where reisomerization into disulfide CysB–CysG is allowed (Supplementary Figure 11c). Hence, we predict that the stiffening effect due to disulfide formation in Ig domains is particularly relevant in tissues containing long titin isoforms, such as skeletal muscles, fetal myocardium and in the heart of species with high proportion of N2BA isoform, like humans[39,41].

Several other posttranslational modifications modulate the mechanical properties of titin. Phosphorylation of both the N2B and PEVK random coil regions changes their persistence length[9,42], resulting in adjustments in the elasticity of titin with maximum effects at low forces. Our simulations show that disulfides in Ig domains can induce changes in titin stiffness of similar magnitude (Supplementary Figure 11d). The stiffening effect of oxidation can be further enhanced by disulfide formation at the N2B random coil region[10]. However, S-glutathionylation, which is a different type of oxidative modification, leads to softening of titin through inhibition of protein folding[3]. Hence, the overall impact of oxidative modifications on titin elasticity depends on the specific residues targeted and the chemical nature of the modifications. To examine experimentally how redox signaling affects the mechanical properties of titin, coordinated efforts involving biochemical identification of oxidative modifications and their impact on titin elasticity at the molecular and cellular levels, together with integrative computational models, will be needed.

Our study reports on the complex crosstalk between the folding/unfolding dynamics of titin's Ig domains and the mechanochemistry of disulfide bonds. This interplay has been recently proposed to be a main driver of the evolution of the titin molecule along the vertebrate lineage[43]. Here, we have identified that disulfide isomerization contributes to the overall mechanical response of titin domains. The relative weight of this mechanism can be inferred form the strong conservation of the triad-forming cysteines (Fig. 6a). Interestingly, other proteins involved in mechanical force production and mechanical support also have arrangements of cysteines and disulfides that can enable force-triggered disulfide isomerization reactions, such as the C10 domain of myosin binding protein C (Supplementary Note 5), or the junctional adhesion molecule A (pdb: 1F97). In the case of titin, the high cost of expression and assembly of this gigantic protein implies a slow turnover and versatile elastic properties. We envision that changes of the mechano-redox environment of myocytes can modify the disulfide bond status of titin via the mechanochemical redox reactions described here, resulting in rapid compensatory regulation of its elasticity. Dysregulation during periods of excessive oxidative stress may contribute to development of heart and/or musculoskeletal diseases[44–46].

## Methods

**Analysis of Ig sequences and structures**. The 151 sequences of Ig domains are automatically annotated in the canonical titin (Uniprot Q8WZ42-1). Nevertheless, a few domains are missing in this annotation. For instance, the best-characterized Ig domain of titin, I91, is not annotated. Such additional domains can be verified by comparing their sequence to other Igs and checking the presence of the strictly conserved tryptophan 35. Accordingly, we manually added 12 Igs to the Uniprot-annotated domains. These domains correspond to positions 2704–2794, 3145–3233, 9581–9670, 12,041–12,133, 12,321–12,411, 12,674–12,765 (I91), 12,854–12,943, 13,030–13,119, 13,837–13,926, 14,319–14,411, 20,618–20,710 and

31,553–31,646 (numbers refer to the starting and final amino acid, according to the numbering of canonical titin Q8WZ42). Following this manual annotation, we report a total of 163 Igs in the canonical human titin (Supplementary Note 3).

**Protein expression and purification**. We used synthetic complementary DNAs (cDNAs) coding for rabbit I65–70 (Genscript) and I69 (Genewiz). Polyproteins $(I69)_8$ and $(I91\Delta Cys)_2$-I69-$(I91\Delta Cys)_2$ were engineered by iterative cloning using *Bam*HI, *Bgl*II and *Kpn*I restriction sites[27]. Final cDNAs were cloned into expression vector pQE80L using *Bam*HI and *Kpn*I (Qiagen). Full protein sequences are provided in Supplementary Note 7. Both I65–I70 and $(I91\Delta Cys)_2$-I69-$(I91\Delta Cys)_2$ were expressed in BLR *E. coli* while the expression of $(I69)_8$ was enhanced by the use of strain RB791[47] (24 h expression at 16 °C). Cultures are induced with 1 mM isopropyl-D-1-thiogalactopyranoside after growth at 37 °C to reach $OD_{600}=$ 0.6–0.8. Extracts from harvested cells are purified in two steps. We use an immobilized metal ion affinity chromatography (IMAC) followed by a size exclusion chromatography in a fast protein liquid chromatography (FPLC) machine using a Superdex 200 10/300 column (GE Healthcare). The polyprotein is eluted in Experimental Buffer (10 mM hepes buffer, pH 7.2, 150 mM NaCl, 1 mM EDTA). Sodium dodecyl sulfate–polyacrylamide gel electrophoresis analysis confirmed size and purity of the samples. Oxidation of the substrates requires an additional step in which the samples eluted from IMAC are incubated overnight in an oxidizing buffer containing 0.3% $H_2O_2$ and 1 mM $CuCl_2$ at 4 °C for $(I69)_8$ and $(I91\Delta Cys)_2$-I69-$(I91\Delta Cys)_2$ and at room temperature for I65–I70. Before the last FPLC step, the samples are incubated during 30–60 min on ice with a large amount of reducing agent (20 mM Tcep). This last step is critical for the AFM experiments. The reduction of the accessible terminal cysteines is required for the covalent attachment to the gold surface while keeping the buried disulfide bonds protected from the solvent. Tcep treatment may reduce some of the preformed disulfide bonds in oxidized I65–I70, which would prevent their detection in the AFM. However, the absence of 10 nm (signature of CysB–CysF), 23 nm (signature of CysF–CysG) and 26 nm (signature of any reduced disulfide) unfolding steps shows that the only disulfide formed in oxidized I69 is CysB–CysG, and that this disulfide is not accessible to Tcep.

**Single-molecule experiments**. We use a customized AFM that sets the force constant by adjusting the piezoelectric positioner according to the deflection of the cantilever[17,48]. A total of 5–20 μl of purified polyprotein solution is absorbed on an ~40 nm gold-coated coverslip. Veeco silicon nitride MLCT cantilevers (Bruker, Camarillo, CA) are mounted on a fluid cell chamber and calibrated using the equipartition theorem[49], obtaining typical spring constants of ~14–17 pN nm$^{-1}$ [17]. We carry out all experiments in Experimental Buffer. We could maintain all protein samples at 4 °C and obtain productive traces within 2–3 weeks following purification. Experimental AFM cycles involve approaching the gold surface to the cantilever to establish protein tethers and retraction until the set force is reached. Cycles run continuously until a tether is formed. Tethers break spontaneously during the experiment or by pulling at high forces at the end of the experimental protocol. AFM traces were selected according to two criteria: (1) for the determination of mechanochemically induced changes in contour lengths of protein domains (Figs 2c, e, 3), we examined single-molecule traces that have at least two steps of the same size; and (2) for the analysis of mechanical stability (Fig. 4), refolding (Figs 2f, 3a and 5) and isomerization rates (Supplementary Figure 2), we selected fingerprinted traces in which all steps can be unambiguously assigned to any of the molecular events described in the text. This second, more stringent fingerprint criterion ensures that the molecular parameters that we measure are not affected by other, rare unrelated events such as nonspecific interactions with the surface or aggregation[17]. For step size assignments, we also considered the modest difference of step sizes due the different applied forces (i.e., 100 and 170 pN) (Supplementary Note 2). The number of observations $N$ corresponds to the total number of unfolding domains summed from different independent traces. The refolding probability (and associated SEM) was calculated with a bootstrap method based on traces that show the same extension in the first and pulse probe (before isomerization)[12]. To measure isomerization rates accurately, we only consider traces that are longer than 20 s, which favor completion of isomerization reactions (Supplementary Figure 3). Around 0.5% of all AFM attempts lead to traces fulfilling these stringent criteria.

**Monte Carlo simulations**. We modeled the molecular elasticity of the I-band of titin from the summed elasticity of the two entropic regions (N2B and PEVK) and the Ig domains using the Freely Jointed Chain model of polymer elasticity[3]. We define the N2B and PEVK springs to be delimited by the amino acid positions 3712–4289 and 10,216–12,022 in the N2BA canonical titin, respectively. In the N2B isoform, the limits of the shorter PEVK region are 11,851–12,022. The number of Ig domains and their cysteine arrangement were obtained from our bioinformatic analysis of the sequence of titin (Supplementary Note 3). The values of contour length and Kuhn length for each region of titin appear in Supplementary Table 6. We considered that in the initial state of the simulations all Ig domains are folded. In the case of oxidized titin, all potential disulfide bonds in Ig domains are formed, while in the reduced titin all cysteines are free. In the simulations, we let the system evolve during 20 min with Monte Carlo steps of 10 ms. We checked that the results

of the simulations do not change by using shorter steps of 1 ms. The force increases and decreases linearly from 0 to 30 pN in a 1 s cycle. All transition rates are retrieved from previous studies or estimated from the experimental parameters measured here (Supplementary Table 7)[2,3].

**Data availability**. All relevant data supporting the findings of this study are available from the authors on reasonable request.

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

## Acknowledgements

J.M.F. acknowledges support from NSF (DBI-1252857) and NIH (GM116122 and HL061228) grants. J.A.-C. acknowledges funding from the Spanish Ministry of Economy, Industry and Competitiveness (MINECO) through grants BIO2014-54768-P and RYC-2014-16604. The CNIC is supported by MINECO and the Pro CNIC Foundation, and is a Severo Ochoa Center of Excellence (MINECO award SEV-2015-0505). D.G. was a recipient of a Marie Sklodowska-Curie Individual Fellowship (656721). We thank Diego Rojas and Georgia Squyres for excellent technical support. We thank Elías Herrero-Galán (CNIC, Madrid) for critical reading of the manuscript.

## Author contributions

D.G., J.M.F. and J.A.-C. designed the research. C.L.B. engineered polyprotein constructs. D.G. produced proteins. D.G. and K.Y. did AFM experiments, structural and bioinformatic analyses. D.G. programmed Monte Carlo simulations. D.G. and J.A.-C. analyzed the AFM data and the results of Monte Carlo simulations. D.G. and J.A.-C. wrote the manuscript with input from all the authors.

## Additional information

**Competing interests:** The authors declare no competing financial interests.

