## [Peer Review File · Nature Communications]

Reviewers' comments:

Reviewer #2 (Remarks to the Author):

The authors have satisfactorily addressed the concerns raised in my initial report and I feel the manuscript is now acceptable for publication.

Reviewer #3 (Remarks to the Author):

The authors present a very thoroughly conducted study describing the appearance and isomerization of disulfide bridges in titin immunoglobulin domains and discuss the effects on molecule elasticity. The results obtained from atomic force microscopy measurements are of high quality and the findings may indeed present an important mechanism to modify the mechanical properties of the titin molecule. The authors are experts in the field of titin Ig-domain modification and with this present study they will make another valuable contribution to the titin research field. However, the manuscript has some weaknesses that should be addressed.

1. The data are presented and discussed to the very detail, which is impressive, but clearly addressed to biochemists familiar with AFM measurements and analysis of intramolecular interactions. I strongly recommend to revise the discussion section in order to increase its comprehensibility and make it accessible and understandable to a more broader oriented non-expert readership.

2. Along this line, the authors should increase their effort to discuss the novel data in the context of previous findings regarding titin modification and also discuss the potential physiological relevance of Ig-domain disulfide bridges (and isomerization) in more detail.

There are several questions left open for discussion:

- The observation that disulfide formation may increase titin stiffness has previously been described by Grützner et al 2009 for the N2-B unique sequence. What impact on titin stiffness could Ig-domain modification have? Is there a way to estimate this by computational modeling as in Grützner 2009?
- Several data from the literature suggest that titin isoform changes can be overruled by changes in its N2-Bus or PEVK phosphorylation status. What impact could Ig-domain modification and changes to Ig-unfolding/refolding rates have in relation to phosphorylation changes? Is there a way to estimate or even test this? This may be of particular importance, since Kötter et al 2016 reported PEVK phosphorylation as the major cause for titin stiffening in response to myocardial ischemia.
- The actual importance of s-glutathionylation versus disulfide formation should be discussed more extensively.
- What experiments should/could be performed to test the functional relevance of disulfide bridges for titin stiffness? I fully understand that it could be beyond the scope of the current study to perform such experiments, but this issue needs to be addressed in the discussion section.

3. In the discussion section the authors point out that the suggested mechanism would be limited mainly to the alternatively splice I-band region of the titin molecule. Does this not limit its relevance for cardiac tissues with high abundance of the shorter N2B titin isoform, especially in mice or rats, but also in human hearts? In the light of a high N2B portion would the myofilament not have to be stretched extensively/unphysiologically to reach unfolding of Ig-domains in the N2BA isoform? However, in skeletal muscles with a predominant expression of N2A isoforms the mechanism could be of particular importance.

NCOMMS-17-20759-T. Reply to reviewers.

Reviewer #3 (Remarks to the Author):

The authors present a very thoroughly conducted study describing the appearance and isomerization of disulfide bridges in titin immunoglobulin domains and discuss the effects on molecule elasticity. The results obtained from atomic force microscopy measurements are of high quality and the findings may indeed present an important mechanism to modify the mechanical properties of the titin molecule. The authors are experts in the field of titin Ig-domain modification and with this present study they will make another valuable contribution to the titin research field.

R: We thank the reviewer for the positive comments about the significance of our work and the quality of our data, and also for the suggestions to integrate our findings in the context of other regulatory mechanisms of titin. As a result, the revised manuscript now includes additional data that illustrate the functional relevance of the new mechanism of protein elasticity based on disulfide oxidation and isomerization.

However, the manuscript has some weaknesses that should be addressed.

1. The data are presented and discussed to the very detail, which is impressive, but clearly addressed to biochemists familiar with AFM measurements and analysis of intramolecular interactions. I strongly recommend to revise the discussion section in order to increase its comprehensibility and make it accessible and understandable to a more broader oriented non-expert readership.

R: To make the discussion more understandable to non-expert readers, we have simplified some parts and included when appropriate additional explanations.

2. Along this line, the authors should increase their effort to discuss the novel data in the context of previous findings regarding titin modification and also discuss the potential physiological relevance of Ig-domain disulfide bridges (and isomerization) in more detail. There are several questions left open for discussion:

R: We have adapted well-established Monte Carlo simulations (Alegre-Cebollada 2014) to understand the physiological relevance of disulfides in the elasticity of titin. The simulations show that disulfides induce a global stiffening of titin, and provide novel insights into the role of disulfides in the context of what is known of the elasticity of titin. To include this new information, we have added the section "Impact of disulfide bonds on the elasticity of titin" to the discussion, a new figure (Supplementary Figure S11) and updated the abstract and the methods section.

- The observation that disulfide formation may increase titin stiffness has previously been described by Grützner et al 2009 for the N2-B unique sequence. What impact on titin stiffness could Ig-domain modification have? Is there a way to estimate this by computational modeling as in Grützner 2009?

R: Our new Monte Carlo simulations show that disulfide bond formation in titin domains leads to global stiffening of the titin filament (new Supplementary Figure S11). As discussed in the section "Impact of disulfide bonds on the elasticity of titin", this effect would add to the reported effect of oxidation of the N2-B unique sequence.

- Several data from the literature suggest that titin isoform changes can be overruled by changes in its N2-Bus or PEVK phosphorylation status. What impact could Ig-domain modification and changes to Ig-unfolding/refolding rates have in relation to phosphorylation changes? Is there a way to estimate or even test this? This may be of particular importance, since Kötter et al 2016 reported PEVK phosphorylation as the major cause for titin stiffening in response to myocardial ischemia.

R: We have compared the effects of phosphorylation of the N2-Bus or PEVK on the elasticity of titin to the stiffening induced by disulfide formation, and found effects can be of the same magnitude (new Supplementary Figure S11d).

- The actual importance of s-glutathionylation versus disulfide formation should be discussed more extensively.

R: This discussion has been included in the last paragraph of the new section “Impact of disulfide bonds on the elasticity of titin” of the discussion.

- What experiments should/could be performed to test the functional relevance of disulfide bridges for titin stiffness? I fully understand that it could be beyond the scope of the current study to perform such experiments, but this issue needs to be addressed in the discussion section.

R: The sort of experiments that would be needed to address this important issue are discussed in the last paragraph of the new section “Impact of disulfide bonds on the elasticity of titin” of the discussion.

3. In the discussion section the authors point out that the suggested mechanism would be limited mainly to the alternatively splice I-band region of the titin molecule. Does this not limit its relevance for cardiac tissues with high abundance of the shorter N2B titin isoform, especially in mice or rats, but also in human hearts? In the light of a high N2B portion would the myofilament not have to be stretched extensively/unphysiologically to reach unfolding of Ig-domains in the N2BA isoform? However, in skeletal muscles with a predominant expression of N2A isoforms the mechanism could be of particular importance.

R: We agree with the reviewer and our simulations now provide data to support the view that disulfide formation is particularly relevant for the elasticity of long titin isoforms. This finding is elaborated in the new section of the discussion “Impact of disulfide bonds on the elasticity of titin”. According to the literature, the proportion of long isoforms in the human heart is 30-50% (Cazorla, 2000; Neagoe 2003), so disulfides are probably also relevant for the mechanical properties of human myocardium. One of our most striking findings is that disulfides increase the unfolding rate of domains by one order of magnitude (Supplementary Figure S4). Hence, the unfolding of oxidized domains can be easily triggered by low, physiological forces in time scales of a few minutes, as shown by our new Monte Carlo simulations (Supplementary Figure S11).

REVIEWERS' COMMENTS:

Reviewer #3 (Remarks to the Author):

Congratulations to this very nice work! I have no further comments.